# Assessing the value of deep neural networks for postoperative complication prediction in pancreaticoduodenectomy patients

**Mikkel Bonde[1], Alexander Bonde[1], Haytham Kaafarani[2], Andreas Millarch[1], Martin Sillesen**[1,3,4]*

**1** Dep. of Organ Surgery and Transplantation, Copenhagen University Hospital, Rigshospitalet, Copenhagen, Denmark, **2** Div. of Trauma, Emergency Surgery and Surgical Critical Care, Massachusetts General Hospital/Harvard Medical School, Boston, Massachusetts, Unites States of America, **3** Center for Surgical Translational and Artificial Intelligence Research, Copenhagen University Hospital, Rigshospitalet, Copenhagen, Denmark, **4** Department of Clinical Medicine, University of Copenhagen Medical School, København, Denmark

* Martin.Sillesen@regionh.dk

**Data Availability Statement:** The de-identified dataset used for this study can be obtained from the authors or the following institutional contacts, provided written authorization from data owners

## Abstract

### Introduction

Pancreaticoduodenectomy (PD) for patients with pancreatic ductal adenocarcinoma (PDAC) is associated with a high risk of postoperative complications (PoCs) and risk prediction of these is therefore critical for optimal treatment planning. We hypothesize that novel deep learning network approaches through transfer learning may be superior to legacy approaches for PoC risk prediction in the PDAC surgical setting.

### Methods

Data from the US National Surgical Quality Improvement Program (NSQIP) 2002–2018 were used, with a total of 5,881,881 million patients, including 31,728 PD patients. Modelling approaches comprised of a model trained on a general surgery patient cohort and then tested on a PD specific cohort (general model), a transfer learning model trained on the general surgery patients with subsequent transfer and retraining on a PD-specific patient cohort (transfer learning model), a model trained and tested exclusively on the PD-specific patient cohort (direct model), and a benchmark random forest model trained on the PD patient cohort (RF model). The models were subsequently compared against the American College of Surgeons (ACS) surgical risk calculator (SRC) in terms of predicting mortality and morbidity risk.

### Results

Both the general model and transfer learning model outperformed the RF model in 14 and 16 out of 19 prediction tasks, respectively. Additionally, both models outperformed the direct model on 17 out of the 19 tasks. The transfer learning model also outperformed the general model on 11 out of the 19 prediction tasks. The transfer learning model outperformed the ACS-SRC regarding mortality and all the models outperformed the ACS-SRC regarding the

American College of Surgeons, National Surgical Quality Improvement Program (ACS-NSQIP) can be obtained. The reason for the current restriction is patient privacy as well as the fact that the ACS-NSQIP does not allow for commercial use of data collected through the ACS-NSQIP. As such, there is a requirement of data use only for specific and approved research projects, and that this is individually approved by the ACS-NSQIP, including through a signed data user agreement. This data user agreement can be obtained by contacting the ACS-NSQIP at baa@facs.org. The American College of Surgeons National Surgical Quality Improvement program (ACS-NSQIP), serving as the governing body for the data used in this study, can be contacted at baa@facs.org. The point of contact is Brian Matel. Contacting researchers will be required to complete a new data user agreement with the ACS-NSQIP if secondary use of the data is desired.

**Funding:** Funded by a grant from the Novo Nordisk Foundation (grant #NNF19OC0055183) to MS. The funders had no role in study design, data collection and analysis, decision to publish, or preparation of the manuscript.

**Competing interests:** Authors AB and MS have founded Aiomic Aps, a healthtech company fielding artificial intelligence models for healthcare use. The present work is for research only and is not related to any commercial activities. This does not alter our adherence to PLOS ONE policies on sharing data and materials.

morbidity prediction with the general model achieving the highest Receiver Operator Area Under the Curve (ROC-AUC) of 0.668 compared to the 0.524 of the ACS SRC.

## Conclusion

DNNs deployed using a transfer learning approach may be of value for PoC risk prediction in the PD setting.

## Introduction

Pancreatic ductal adenocarcinoma (PDAC) is a leading cause of cancer-related deaths in Western countries, with a 5-year survival rate of approximately 12%, making it the cancer with the lowest 5-year survival rate in the United States [1]. Furthermore, patients with successfully resected tumors have a 3-year survival rate of only 20–34%, [2] which is attributable to a combination of aggressive tumor growth patterns and poor response to oncological treatment [3–5].

Surgical resection in the form of pancreaticoduodenectomy (PD, Whipple's procedure), distal pancreatectomy (DP) or total pancreatectomy (TP)—provides the only curative option for patients with PDAC. These procedures are, however, associated with a plethora of postoperative complications (PoCs) such as Superficial Surgical Site Infections (SSSIs), Organ/Space Specific Surgical Site Infections (OSSI), venous thromboembolism (VTE's), hemorrhage, and death, collectively affecting upwards of 40% of patients undergoing PD [6]. These complications not only prolong the surgical treatment phase and subject patients to significant morbidity, but could furthermore render the patient unable to proceed with adjuvant chemotherapy due to frailty issues [7].

Due to these factors as well as the fact that upwards of 80% of successfully resected patients suffer tumor recurrence [8], the risk of an operative approach to PDAC treatment needs to be carefully weighed against the potential benefits, and tools for identifying PoC risks thus play an important role in selecting optimal treatment strategies for PDAC patients. While multiple risk prediction tools have been proposed for both general surgical and PDAC patients, these have reported varied performance in terms of identifying PoC risks [9, 10], especially for PD patients [11]. Novel approaches such as artificial intelligence (AI) deep neural networks (DNNs) have, however, recently shown superior performance over legacy approaches in PoC prediction [12] although the potential value for pancreatic surgery patients remain unknown.

Legacy risk prediction models often have the inherent drawback of being trained on general surgical cohorts with subsequent applications to specific surgical procedure such as PD. In contrast, DNN approaches can leverage the power of transfer learning [13], meaning that the model can be trained and learn general features from one patient cohort (e.g., learning features associated with PoC's in general surgical populations) with subsequent retraining and fine tuning on a specific surgical procedure or cohort (e.g., patients undergoing a PD procedure. This makes it possible to create new models with knowledge of previously predicted input and output relationship which can then further be refined and improved through training on new data [14]. We have previously demonstrated the value of this approach [13]. Investigating whether DNNs and the potential of transfer learning could be of value in predicting PoC risk for PD patients presents the goal of this study. We hypothesize that transfer learning of a DNN previously trained on a large-scale dataset with many surgical procedure types, to a dataset consisting exclusively of PD patients could be superior to legacy approaches including the

American College of Surgeons Surgical Risk Calculator (ACS-SRC) [15] for task including predicting the 30-day risk of mortality and morbidity as defined by the National Surgical Quality Improvement Program (NSQIP) [16].

## Methods

This study was conducted using a dataset obtained from the US American College of Surgeons (ACS) NSQIP, which includes manually curated PoC's from more than 700 US hospitals across 2,941 different procedure subtypes. For this study we used the 2002 to 2018 dataset available through NSQIP. The study and the use of the dataset was approved by NSQIP. IRB approval was waived by the Massachusetts General Hospital IRB, including the requirement for informed consent.

Data was accessed May 3rd, 2023. This was a purely retrospective study on data obtained from the NSQIP quality registry. As such, no interaction with patients was required. The study group did not have access to data enabling the identification of individual participants, and patient consent was thus neither required (as per the IRB decision to waive the requirement for consent), nor possible due to the de-identified nature of the dataset.

### Datasets and modelling approaches

The dataset included manually labelled data from 5,881,881 million patients with more than 150 different variables for each patient such as perioperative biochemistry, height, weight, age, smoking status, comorbidities, demographic, American Society of Anesthesiology (ASA) score, and postoperative complications as defined by NSQIP [16] and shown in Table 1.

A graphical representation of patient selection and allocation into training and test data is illustrated in Fig 1. Of the 5,881,881 patients in the NSQIP dataset, we identified 31,944 patients as having undergone a PD procedure (as indicated by the CPT codes 48150, 48152, 48153, and 48154). A total of 216 patients were excluded from the study due to having undergone a PD operation with a duration of 2 hours or less because we assessed that the procedure may not have been completed and there would thus be a risk of incorrect coding, resulting in a final sample size of 31,728 patients.

The dataset of 31,728 PD patients was randomly split into two datasets: a dataset with 12,907 PD patients (40%) which would be recombined with the non-PD dataset (5,862,034 patients) containing non-PD patients and thus all operation types (termed "general dataset" in the following), which was fielded to allow the model to learn both general and PD-specific features. The second dataset was a dedicated PD dataset containing the remaining 19,037 (60%) of patients (PD dataset): This dataset contained only PD patients.

The overall objective of the modeling approaches was to predict 30-day risk of death and/or the occurrence of 18 different postoperative complications as defined by NSQIP [16] using a DNN and Random forest for multi-label classification enabling the prediction of all outcomes by a single model (please see prediction variables below). To align with the ACS-SCR, we further included a target termed "morbidity", defined as the occurrence of any of the 18 NSQIP-defined complications within a 30-day period following surgery.

The datasets were used for training and testing four different multi-label modelling approaches, aiming at identifying the optimal training and dataset use approach in the PD setting:

1. Training of a DNN on the general dataset with direct porting and testing on the PD dataset (general model)

**Table 1. The incidence of postoperative complication (prediction variables) for the three datasets before the split into validation/training sets are depicted above with the number of patients experiencing each variable labelled.** PD: Pancreaticoduodenectomy.

| | General dataset (n = 5,874,941) | PD dataset (n = 17,037) | Test dataset (n = 2,000) |
|---|---|---|---|
| **Superficial Surgical site infection** | 79,986 (1.4%) | 1,348 (7.9%) | 153 (7.7%) |
| **Deep surgical site infection** | 21,314 (0.4%) | 345 (2.0%) | 38 (1.9%) |
| **Organ/space surgical site infection** | 49,259 (0.8%) | 2,321 (13.6%) | 247 (12.4%) |
| **Wound disruption** | 21,921 (0.4%) | 242 (1.4%) | 23 (1.2%) |
| **Postoperative pneumonia** | 54,850 (0.9%) | 704 (4.1%) | 86 (4.3%) |
| **Unplanned intubation** | 43,288 (0.7%) | 716 (4.2%) | 77 (3.9%) |
| **Pulmonary embolism** | 18,967 (0.3%) | 193 (1.1%) | 32 (1.6%) |
| **Ventilator dependence >48 hours** | 43,476 (0.7%) | 582 (3.4%) | 66 (3.3%) |
| **Progressive renal insufficiency** | 14,349 (0.2%) | 141 (0.8%) | 13 (0.7%) |
| **Acute renal failure** | 15,511 (0.3%) | 183 (1.1%) | 22 (1.1%) |
| **Urinary tract infection** | 62,844 (1.1%) | 514 (3.0%) | 59 (4.0%) |
| **Stroke** | 11,256 (0.2%) | 51 (0.3%) | 5 (0.3%) |
| **Cardiac arrest** | 17,383 (0.3%) | 211 (1.2%) | 19 (1.0%) |
| **Myocardial infarction** | 21,027 (0.4%) | 193 (1.1%) | 21 (1.1%) |
| **Deep vein thrombosis** | 32,230 (0.6%) | 478 (2.8%) | 59 (3.0%) |
| **Sepsis** | 43,464 (0.7%) | 1,271 (7.5%) | 158 (7.9%) |
| **Septic shock** | 23,498 (0.4%) | 565 (3.3%) | 70 (3.5%) |
| **Bleeding requiring transfusion** | 303,726 (5.2%) | 3,593 (21.1%) | 412 (20.6%) |
| **Death** | 57,605 (1.0%) | 349 (2.0%) | 40 (2.0%) |

2. Transfer learning of a DNN, from a general to a PD specific setting. This model included training on the general dataset, with subsequent transferring to the PD dataset for retraining (transfer learning model)

3. Direct training of a DNN only on the PD dataset (direct model)

4. Direct training of a Random Forrest (RF) model directly on the PD dataset, serving as benchmarking (RF model)

Finally, models were benchmarked against the American College of Surgeons (ACS) Surgical Risk Calculator (SRC), where a mortality risk as well as a compound morbidity risk value was included in the NSQIP dataset.

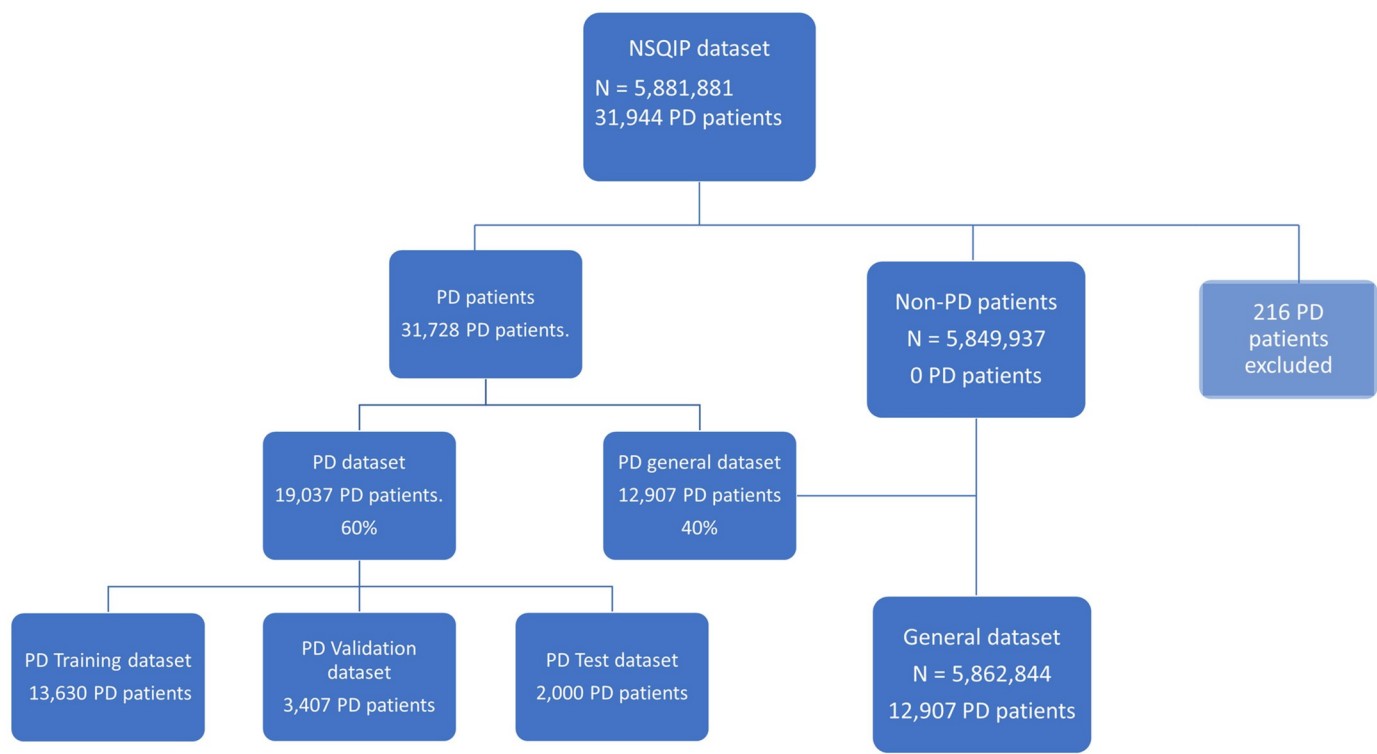

**Fig 1. 5.881.881 patients were in the National Surgical Quality Improvement Program (NSQIP) dataset, 31.944 of whom were PD patients.** 216 of these patients were excluded because of an operation time of less than 120 minutes. The remaining 31.728 patients were split into two datasets. One dataset with 40% of the PD patients which was recombined the with the patients from the remainder of the NSQIP dataset (General dataset and the second data frame which was the 60% were split into a training set, validation set and a test set that was used after the training of all the models to test their accuracy.

To separate the PD patients between training, validation, and test sets, we randomly selected 2,000 PD patients (6.3%) into a test set to ensure the validity of all the models after training. The remaining 17,037 PD patients were then randomly split into an 80% training set (consisting of 13,632 PD patients) and a 20% validation set (consisting of PD 3,407 patients) which was used to test validity and tune hyperparameters for modelling approaches 1–4.

## Model architecture

The model architecture is depicted in Fig 2. Categorical values were converted into integers, and a dimension was assigned to each category in an embedding matrix. To determine the dimension, we multiplied the cardinality (the number of unique values) of the variable by 1.6 and raised it to the power of 0.56. The resulting value was compared to a dimensional space of 600 used from the fast.ai library [17], and the lower value was selected as the dimension of the given category [18]. This process was repeated for all categorical variables, and the resulting embeddings were passed into the same embedding space with enough dimensions to include separate dimensions for each categorical variable.

The categorical variables were then passed through a dropout layer, followed by a normalization layer with the continuous variables. This was followed by a linear layer and a rectified linear unit (ReLU) activation function layer. The resulting tensor was passed through a subsequent normalization, dropout, linear, ReLU activation function, normalization, and another dropout layer before finally passing through a linear layer with 19 output variables denoting

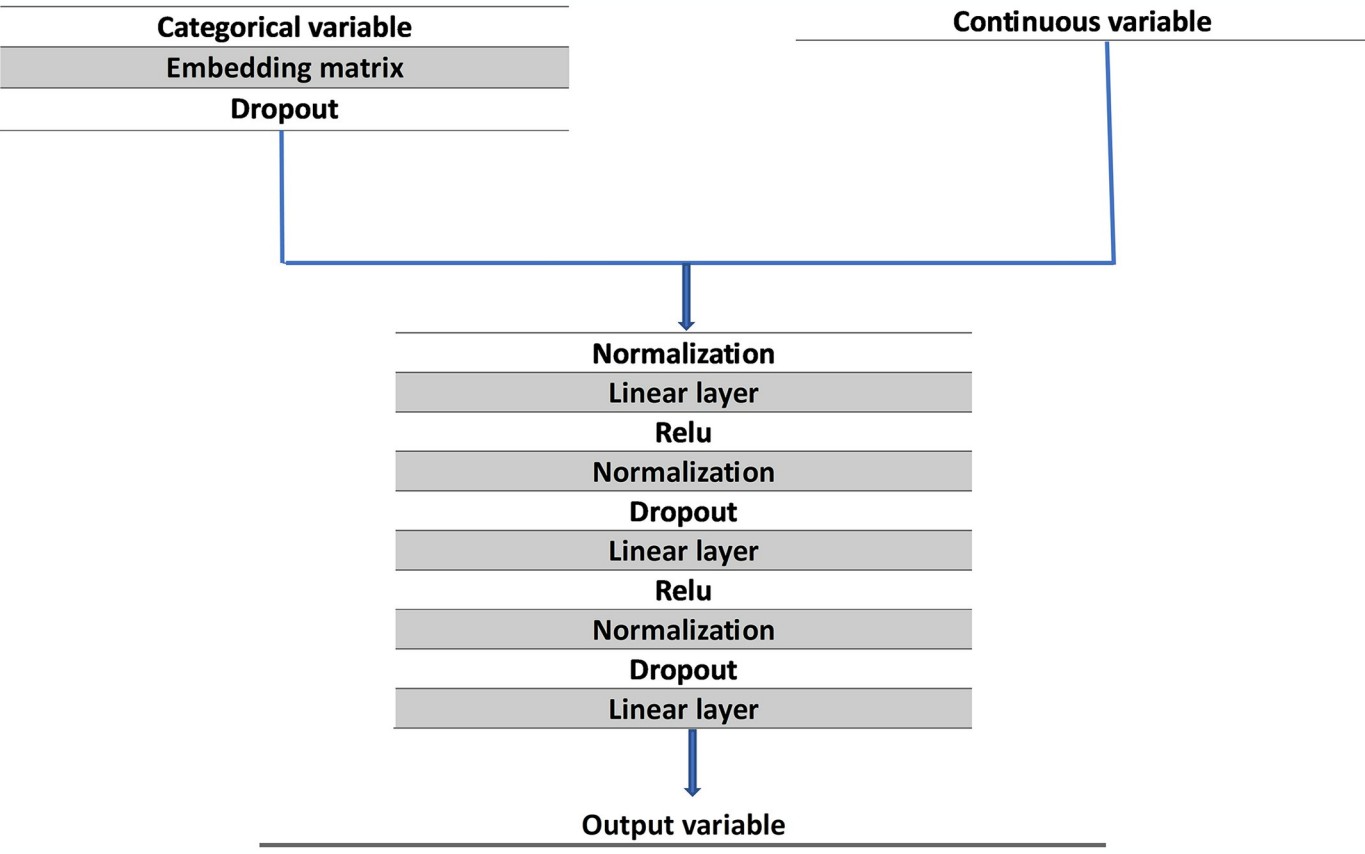

**Fig 2. Model architecture with all layers depicted.**

NSQIP's 18 different complications and death. Backpropagation was used to train the trainable parameters through the Adam optimizer and combined with the loss function flattened Binary Cross Entropy with Logits Loss with positive weights.

To counteract the imbalance in the data, positive weights were applied to the loss function of all DNNs since there were fewer positive outputs than negative ones. This method increases the impact of the minority class on gradient updates during the training process by multiplying the loss with the positive weights when the minority class is misclassified. The positive weight for each output variable was determined by calculating the ratio of negative outcomes to positive outcomes for each variable in the training sets [19].

The DNNs all had 53 embedding layers and trained for 5 epochs, but they differed in terms of the number of trainable parameters, learning rates, weight decay, and weights in the loss function. The general model was trained on a neural network model with 1,253,130 trainable parameters, a learning rate of 3e-3, and a weight decay of 0.2. The transfer learning model also had 1,253,130 trainable parameters, a learning rate of 2e-4, and a weight decay of 0.2. The direct model trained on PD patients had 703,219 trainable parameters, a learning rate of 2e-4, and no weight decay was specified for this model.

To compare the DNNs with a conventional method of handling structured tabular data we created a random forest model. Our random forest consisted of 100 trees, each trained on a sample of 75% of the total data. We used the DecisionTreeClassifier from Scikit-learn with a minimum of five samples per node to train each tree. For each split point in the decision trees,

we randomly sampled 50% of the columns. We set the minimum number of samples required to be a leaf node to 40.

## Input variables

The input variables for the models included 64 different factors, which were gathered preoperatively as well as operation time which was recorded after the operation. These variables can be classified as either continuous or categorical. Continuous variables are hierarchical numerical values, such as weight, height, age, and protein levels in the blood, while categorical variables include smoking status, comorbidities, type of anesthesia used, specialty under which the patient was treated, and more.

Continuous input variables were standardized by mean and standard deviation (Z-score normalization). Missing continuous data were handled firstly by replacing the missing value with the median value for the corresponding group and secondly, for each variable where missing values were imputed, generating a new binary categorical variable. This binary variable indicates the presence or absence of missing data for each observation, effectively signaling whether the original continuous variable value was missing for each patient. If any categorical variables had missing data, they were given their own separate category and included in the dataset. Both the categorical and continuous variables were then converted into vectors and updated in each epoch of training via the embedding process. An overview of the input data can be found in S1 Table in S1 Data.

## Prediction variables

The prediction variables in the model were the PoC's as defined by NSQIP, occurring up to 30 days after the surgical procedure. Of these, 18 were related to morbidity and the last was mortality. To evaluate the performance of the models, the Area under the receiver operating characteristic curve (ROC-AUC) metric was used on the test set of 2,000 PD patients. This metric was calculated for each of the 19 output variables (mortality and 18 different postoperative complications) across all four models. The average ROC-AUC value was then determined for the 18 morbidity variables in each of the four models. This value was compared to that of the risk calculator, which only provided the probability of morbidity and mortality without specifying the type of morbidity. For the prediction variables, the dataset contained no missing data.

## SHAP

We calculated Shapley values with the 'SHAP' library [20] to evaluate and visualize the impact of each input variable on the predictions made by our deep neural network (DNN) models. This method assesses the contribution of each input variable to the predicted outcome on the test set. The underlying principle is based on cooperative game theory. The Shapley value represents the average contribution of a feature to the prediction outcome, taken across all possible combinations of input features. This approach allows for a comprehensive understanding of how each input variable influences the models' predictions [20].

## Model implementation

The models in this study were developed using Python 3.7.6, with PyTorch 1.13.1 [21] and fastai 2.7.12 [17]. Performance metrics were derived using scikit-learn 0.22.1 [22].

## Results

The performance of the models on the test data is depicted in Table 2 and Fig 3 with the ROC_AUC as the evaluation metric for all 19 variables. The direct model (trained and tested on PD data only) generally had the poorest performance among the four models only having the best performance for predicting myocardial infarction (MI). The general model (trained on the general dataset containing both general and PD patients) and the transfer learning model (trained on the general dataset with subsequent retraining on the PD dataset) outperformed it on 17 out of the prediction tasks while the RF Model outperformed it on 14 out of the 19 prediction tasks.

Furthermore, Table 2 and Fig 3 demonstrates that the best performing models were the general model as well as the transfer learning model. As is depicted in Fig 2B and 2E respectively, the transfer learning model outperformed the RF model in 16 out of the 19 prediction tasks and the general model outperformed it in 14 out of the 19 predictions tasks. When comparing the general model with the transfer learning model as shown in Fig 3C they exhibited similar performances on most of the outputs, with the transfer learning model outperforming the general model in 11 out of the 19 prediction tasks.

The ACS-SRC only report two variables to the NSQIP dataset: morbidity risk and mortality risk. To compare the performance of the four models with the ACS-SRC, the average ROC_AUC of all variables, except the deceased variable, was calculated and labeled as "morbidity" in Table 3. This table demonstrates that all models outperformed the risk calculator regarding the morbidity risk with the general model achieving the highest average morbidity risk ROC_AUC of 0.679. The transfer learning model and RF model also outperformed the ACS-SRC when assessing the morality risk however, all models faired similar with the results

**Table 2. The overall performance of the four models on all variables in the test set with Receiver Operator Characteristics Area Under the Curve (ROC_AUC) values as the metric.** The general model was trained on a general surgery patient cohort, the transfer learning model was trained on the general surgery patient cohort and transferred to a PD-specific patient cohort, the direct model, and the Random Forest model (RF) was trained exclusively on the PD-specific patient cohort.

| Complication | General model | Transfer learning model | Direct model | RF model |
|---|---|---|---|---|
| Superficial Surgical site infection | 0.608 | 0.582 | 0.537 | 0.575 |
| Deep surgical site infection | 0.695 | 0.706 | 0.622 | 0.593 |
| Organ/space surgical site infection | 0.580 | 0.581 | 0.517 | 0.608 |
| Wound disruption | 0.676 | 0.702 | 0.630 | 0.577 |
| Postoperative pneumonia | 0.639 | 0.629 | 0.562 | 0.567 |
| Unplanned intubation | 0.642 | 0.664 | 0.634 | 0.663 |
| Pulmonary embolism | 0.662 | 0.683 | 0.615 | 0.651 |
| Ventilator dependence >48 hours | 0.706 | 0.705 | 0.669 | 0.701 |
| Progressive renal insufficiency | 0.648 | 0.676 | 0.590 | 0.638 |
| Acute renal failure | 0.718 | 0.750 | 0.713 | 0.763 |
| Urinary tract infection | 0.726 | 0.687 | 0.623 | 0.665 |
| Stroke | 0.738 | 0.615 | 0.714 | 0.583 |
| Cardiac arrest | 0.643 | 0.685 | 0.646 | 0.610 |
| Myocardial infarction | 0.601 | 0.636 | 0.657 | 0.557 |
| Deep vein thrombosis | 0.653 | 0.634 | 0.578 | 0.603 |
| Sepsis | 0.645 | 0.626 | 0.552 | 0.641 |
| Septic shock | 0.695 | 0.713 | 0.629 | 0.702 |
| Bleeding requiring transfusion | 0.761 | 0.753 | 0.733 | 0.738 |
| Death | 0.648 | 0.678 | 0.660 | 0.672 |

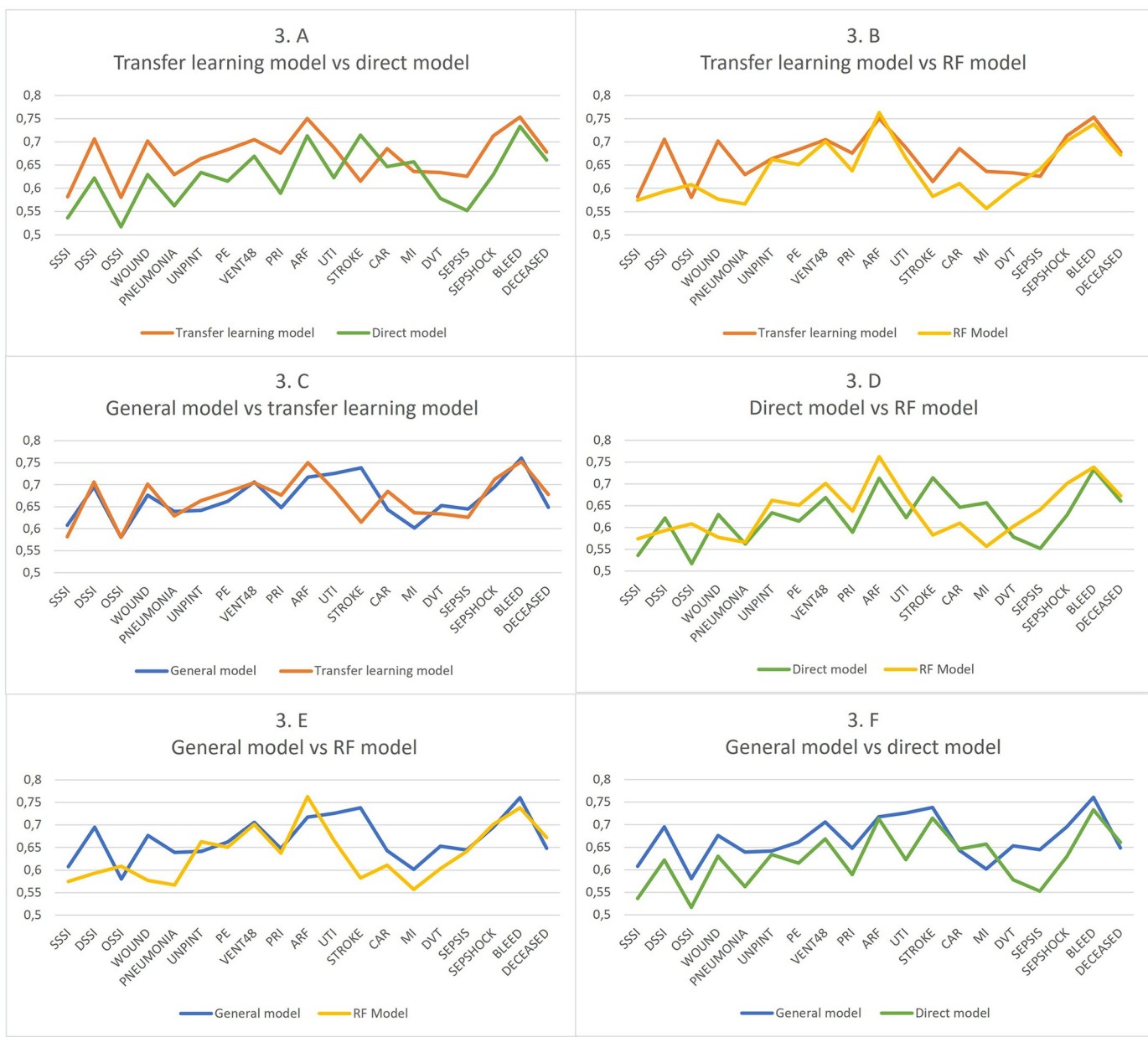

**Fig 3. Performance metrics measures as Receiver Operator Characteristics Area Under the Curve (ROC AUC) of the four different modelling approaches benchmarked against each other for predicting the mortality and the 18 different complications included in the National Surgical Quality Improvement Program (NSQIP) dataset.** SSSI: Superficial Surgical site infection, DSSI: Deep surgical site infection, OSSI: Organ/space surgical site infection, WOUND: Wound disruption, PNEUMONIA: Postoperative pneumonia, UNPINT: Unplanned intubation, PE: Pulmonary embolism, VENT48: Ventilator dependence >48 hours, PRI: Progressive renal insufficiency, ARF: Acute renal failure, UTI: Urinary tract infection, STROKE: Stroke, CAR: Cardiac arrest requiring CPR, MI: Myocardial infarction, DVT: Deep vein thrombosis, SEPSIS: Sepsis, SEPSHOCK: Septic shock, BLEED: Bleeding requiring transfusion, DECEASED: Mortality.

**Table 3. The average morbidity Receiver Operator Characteristics Area Under the Curve (ROC_AUC) scores of the four models, calculated on the test set.** Additionally, the table includes the average morbidity and mortality risk scores obtained from the same test set derived from the American College of Surgeons Surgical Risk Calculator (ACS-SRC).

| Complications | General model | Transfer learning model | Direct model | RF-model | ACS-SRC |
|---|---|---|---|---|---|
| **Morbidity** | 0.669 | 0.668 | 0.623 | 0.635 | 0.524 |
| **Mortality** | 0.648 | 0.678 | 0.661 | 0.672 | 0.667 |

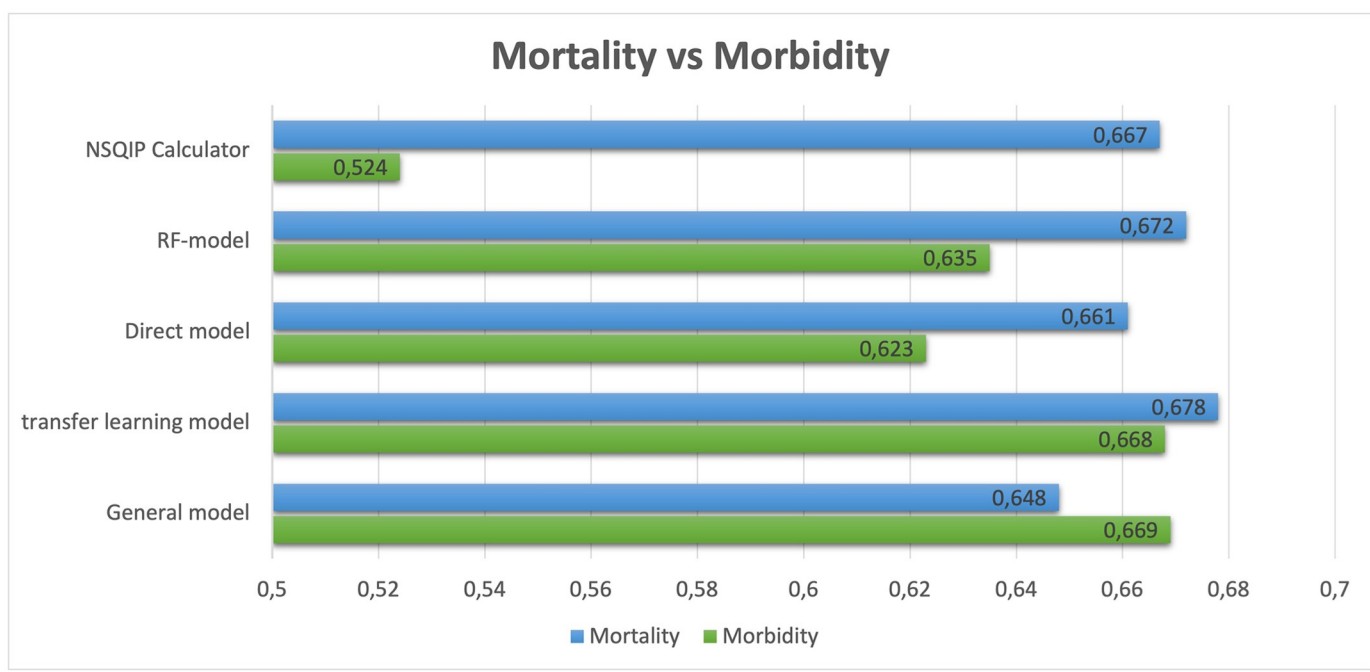

**Fig 4. The average morbidity score and mortality Area Under the Receiver Operator Curve (ROC AUC) of the 4 models as well as the American College of Surgeons Surgical Risk Calculator (ACS-SRC).**

ranging from the lowest from the general model of 0.648 to the best of the transfer learning model with a result of 0.678 (Fig 4).

Considering 'SHAP' values for the transfer learning model and the direct model, as shown in Figs 5 and 6, the models' predictions are based on different variable interactions. For example, "Serum albumin" was the fourth biggest driver of complications in the direct model but the thirteenth biggest driving factor in the transfer-learning model. In contrast, "gender", which is not ranked within the top 40 in the direct model, was the fourth most significant driving factor in the transfer-learning model. Furthermore, the transfer-learning model, seem to have more driving factors compared to the direct model, which indicates that the transfer-learning model has a higher complexity. Even though the models have some differences in which value ranked the highest, there was a significant overlap in which values were generally regarded as the most important. These values included "Weight", "Age" and "Operation time" which were top 3 driving factors in both models.

## Discussion

In this study, we demonstrate that transfer learning of a DNN (transfer learning model) as well larger models trained on a diverse range of operations (general model) outperform alternative approaches such as models using random forest (RF model), DNN exclusively trained on PD patients (direct model) and non-deep learning AI models (ACS-SRC) when risk-predicting post-operative complications in PD patients. The DNN transfer learning approach thus outperformed the ACS-SRC, suggesting a potential value of using this approach when targeting limited-volume operation subtypes such as PD where direct transfer of pretrained models on large PD datasets or direct de-novo training of risk prediction models may not be feasible.

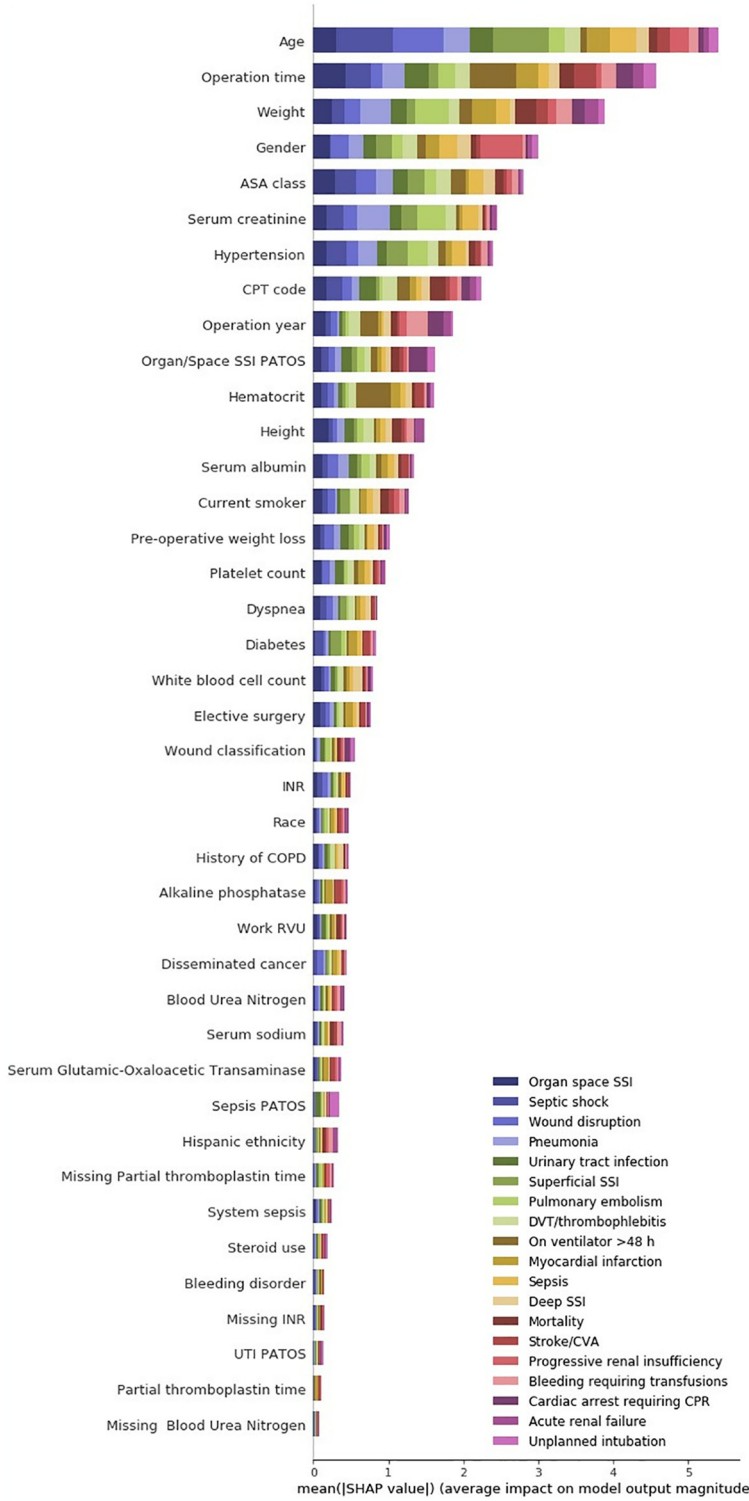

**Fig 5. SHAP values for the transfer learning model.** The x-axis contains the average impact on the color-coded prediction tasks and the y-axis represent the input variables hierarchically dependent on impact level. PATOS: Present at time of surgery.

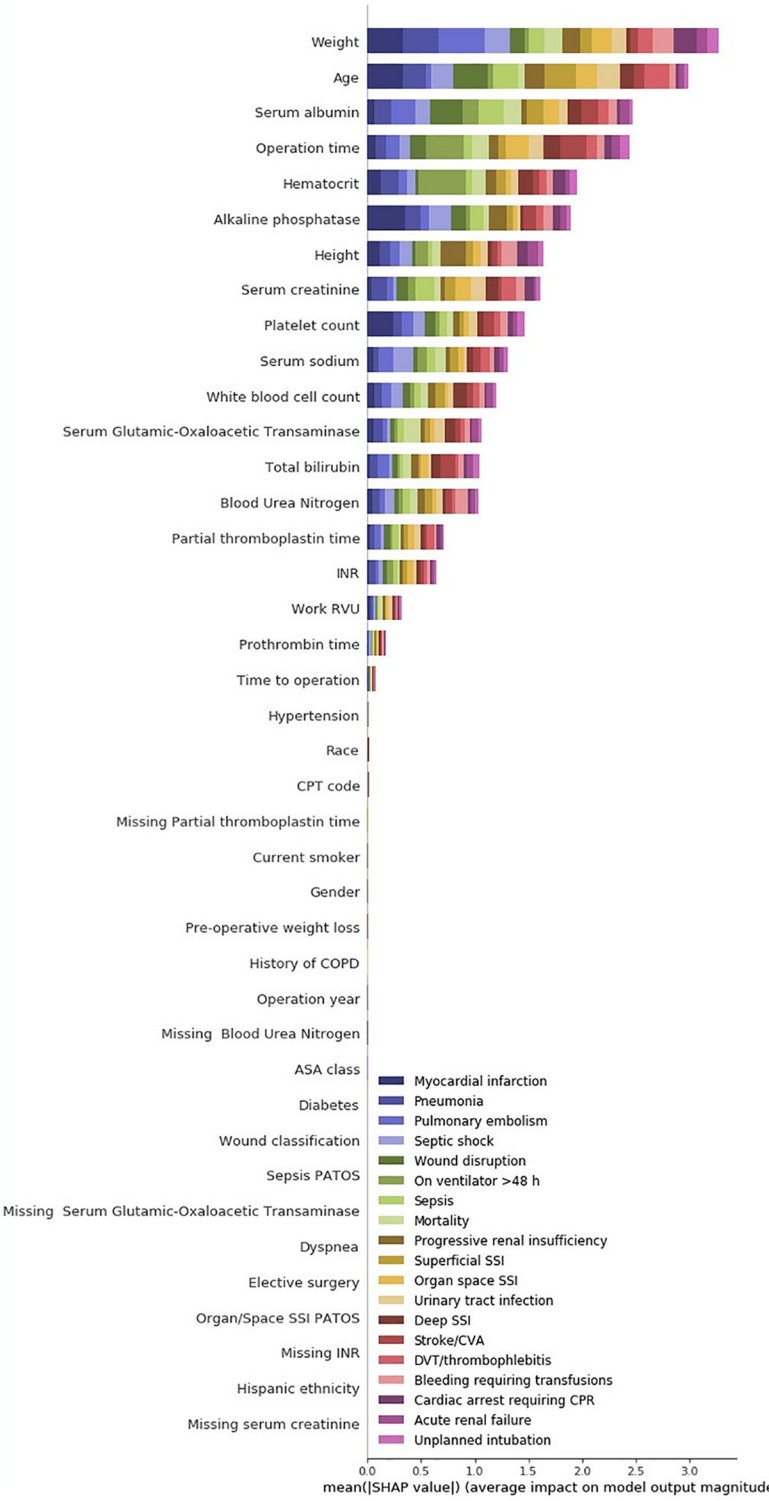

**Fig 6. SHAP values for the direct model.** The x-axis contains the average impact on the color-coded prediction tasks and the y-axis represent the input variables hierarchically dependent on impact level. PATOS: Present at time of surgery.

By utilizing DNNs that capture relationships between input and output variables of common diseases and treatment options, it is possible to develop, coherent predictive models even with limited data available for limited surgical procedures. By leveraging transfer learning techniques, it becomes possible to augment the size of small datasets by leveraging the knowledge acquired from other and potentially larger datasets. This approach expands the amount of available data for training larger models, thereby offering a potential for improving outcomes in the PD setting.

Comparing our study's findings with the results from deploying the ACS-SRC on PDs for neuroendocrine tumors by Dave et al. [23], both the general and the transfer learning model exhibited superior predictive performance in comparison to the NSQIP calculator. Our study incorporated a comparable variable termed 'morbidity,' aligning with the 'serious complication' variable examined by Dave et al. (Table 3). Notably, Dave et al. reported an AUC of 0.55 for their 'serious complication' value, whereas both our general and transfer learning models achieved an AUC of 0.67. However, it should be acknowledged that while there were overlapping complications, there existed differences in the specific variables included in our respective studies, making it difficult to directly compare these two.

In a study from Aoki et al. [6] an attempt was made to predict a value referred to as 'serious morbidity' resulting in an ROC AUC of 0.708. However, the definition of 'serious morbidity' in this study was based on the presence of a Clavien–Dindo classification grade of IV or V, which introduces a disparity between this study and ours in terms of prediction variables. Similarly, a study from Braga et al. [24] also predicted major complications and achieved a ROC AUC of 0.743. However, their definition of major complications also aligned with a Clavien-Dindo classification of IV or V and included a variety of other types of complications, making it challenging to compare their study directly with ours.

As such, in assessing the performance of this model versus previous approaches, it is important to underline that prediction targets are often not aligned.

Most previous approaches have targeted PD specific complications such as pancreatic fistula development, which is indeed a major driver of postoperative complications in the PD setting. In contrast, this study focusses on the prediction of general non-PD specific complications. The rationale behind this choice stems from the fact that key drivers of fistula development (pancreatic texture and pancreatic duct diameter) are often not available for risk prediction before the time of surgery and models incorporating these features are thus of little use in the preoperative setting where the decision on whether to proceed with an operative strategy must be made.

The relevance of fistula development as a driver of other complications should, however, not be underestimated. Fistula development is a recognized driver of other complications, including SSI's [25] and postoperative hemorrhage [26]. Previous results fielding a large-scale postoperative risk prediction model using a DNN approach on multiple surgical procedure subtypes from the NSQIP dataset, yielded a combined morbidity risk prediction ROC AUC of 0.87 [12], which is superior to the combined morbidity risk prediction ROC AUC of 0.678 demonstrated here. The reason behind this suboptimal performance is likely multifaceted but could include that fact the fistula development risk did not factor into risk calculations in this model. The fact that the DNN approach presented here is on-par with or superior to previous approaches does, however, highlight the fact that even with state-of-the art DNN approaches that have previously demonstrated superior performance compared with legacy approaches [12], PD risk prediction continues to a difficult task where models exhibiting excellent performance still remain elusive. Future efforts could potentially benefit from incorporating methods for assessing fistula risk by including preoperatively available data points assessing pancreatic

texture and duct diameter, potentially through automated density analyses of preoperative CT scans combined with pancreatic duct diameter measurements.

This study has limitations that should be acknowledged. As is the case for all studies utilizing registry datasets, models are dependent on the quality and transferability of the data which the model is built upon. As an example, temporal information on when during treatment data points are obtained cannot be assessed. This poses a challenge especially regarding continuous variables such as biochemistry, which are susceptible to fluctuation depending on the time of measurement. A second limitation is that the PD dataset is of limited size, although this was also the rationale for assessing the value of transfer learning approaches in the first place. The PD dataset was used for training all but the general model as well as validation and testing of all models. Therefore, the limited size of this dataset reduces the generality of the findings as well as hinder the learning of very complex relationships between variables. The limited size of the PD dataset is particularly challenging for rare outcome predictions such as stroke which, in the test set, only occurs with 5 patients. A third limitation of this study could be attributed to the underlining patient demographics and treatment strategies. NSQIP contains data primarily from US patients, and it thus cannot be assessed how this model would perform on non-US patients or hospital systems. Furthermore, it should be noted that although the ability of DNNs to include a multitude of relevant input variables offers the approach a position of strength over conventional regression-based approaches, this often also hinders manual use of the model as it is impractical for the clinical user to input several hundred parameters to the model for each risk prediction. Ideally, actual clinical use of DNNs would thus require automated embedding of the DNNs directly into the electronic health record (EHR) systems. Lastly it is worth noting that as with all studies concerning DNNs, the black box issue of which factors the model perceives as most relevant, is still an unsolved problem. Therefore, it is difficult for the model to address the rationale behind specific predictions hindering the ability to determine the relationships between variables which the model found most important. We have attempted to try and visualize the importance of the most important variables using SHAP values. However, visualizing the importance of a single feature in a non-linear model-still presents a significant challenge.

Even with these limitations, we conclude that DNNs and transfer learning approaches may have a value in predicting general complications in the setting of low-volume surgical cases such as PDs, although overall performance improvements and EHR system integrations are likely needed before models can see actual clinical use.

## Supporting information

**S1 Data. Supplementary data-2.**
(DOCX)

## Author Contributions

**Conceptualization:** Mikkel Bonde, Alexander Bonde, Andreas Millarch, Martin Sillesen.

**Data curation:** Mikkel Bonde, Alexander Bonde, Martin Sillesen.

**Formal analysis:** Haytham Kaafarani, Andreas Millarch, Martin Sillesen.

**Funding acquisition:** Martin Sillesen.

**Investigation:** Mikkel Bonde, Alexander Bonde, Andreas Millarch, Martin Sillesen.

**Methodology:** Mikkel Bonde, Alexander Bonde, Andreas Millarch, Martin Sillesen.

**Project administration:** Martin Sillesen.

**Software:** Alexander Bonde, Andreas Millarch.

**Supervision:** Haytham Kaafarani.

**Validation:** Haytham Kaafarani.

**Writing – original draft:** Mikkel Bonde.

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
