## [Decision Letter · Decision Letter 0]

30 Jan 2024

PONE-D-23-30439ASSESSING THE VALUE OF DEEP NEURAL NETWORKS FOR POSTOPERARTIVE COMPLICATION PREDICTION IN PANCREATICODUODENECTOMY PATIENTSPLOS ONE

Dear Dr. Sillesen,

Thank you for submitting your manuscript to PLOS ONE. After careful consideration, we feel that it has merit but does not fully meet PLOS ONE’s publication criteria as it currently stands. Therefore, we invite you to submit a revised version of the manuscript that addresses the points raised during the review process.

Please revise.

We look forward to receiving your revised manuscript.

Kind regards,

Academic Editor

PLOS ONE

Journal Requirements:

3. Thank you for stating the following in the Competing Interests section: "I have read the journal's policy and the authors of this manuscript have the following competing interests: Authors AB and MS have founded Aiomic Aps, a healthtech company fielding artificial intelligence models for healthcare use. The present work is for research only and is not related to any commercial activities."

Reviewers' comments:

Reviewer's Responses to Questions

**Comments to the Author**

1. Is the manuscript technically sound, and do the data support the conclusions?

Reviewer #1: No

Reviewer #2: Yes

2. Has the statistical analysis been performed appropriately and rigorously? 

Reviewer #1: No

Reviewer #2: Yes

3. Have the authors made all data underlying the findings in their manuscript fully available?

Reviewer #1: Yes

Reviewer #2: Yes

4. Is the manuscript presented in an intelligible fashion and written in standard English?

Reviewer #1: Yes

Reviewer #2: Yes

5. Review Comments to the Author

Reviewer #1: I appreciate the opportunity to examine this interesting paper. Nevertheless, there are a few concerns regarding the work.

Conduct a comprehensive review of existing literature pertaining to the utilization of machine learning and deep learning techniques in the context of mortality and morbidity, as well as any relevant studies in related fields. The utilization of Deep Neural Networks (DNN) and Random Forest (RF) can be justified based on their respective advantages and capabilities. There are numerous AI algorithms available, with the latest one being ensemble learning, which offers superior performance. Deep neural networks (DNN) may not consistently perform as effectively as other machine learning methods, but they are particularly notable for their application in image processing. This is evident when the area under the curve (AUC) has a relatively low value. A value that is considered acceptable is 0.75 or higher.

The authors should explicitly state their objective and anticipated outcome in the objective section. As I comprehend, there are multiple morbidity outcomes and death outcomes, and the authors conducted different modeling for each result. Alternatively, is it a prediction involving many classes? The authors should explicitly articulate this.

In the data section, specify the percentage of missing values for final variables and identify the data imputation method employed.

The authors determined the final features from the 150 variables listed in the study. The ultimate result must also be explicitly expressed. It is necessary to provide a clear explanation of how the dataset is allocated to the four models. How did the authors address the issue of data imbalance, which can explain the low AUC value? There is a noticeable imbalance in the data when merging patients with PD with people without PD.

Summary statistics of each dataset are compared against the outcome using chi square analysis, yielding the chi square value for categorical and corresponding p-value continuous variable.

Did the authors undertake parameter optimization on the RF model to enhance its performance? Support Vector Machines (SVM) are a suitable option for conducting benchmarking.

I suggest incorporating Net Index Reclassification (NRI) for the purpose of comparing it with traditional methodologies.

Additionally, the use of SHAP can provide a clearer understanding of the relationship between input properties and the resulting conclusion.

Reviewer #2: I commend you for taking up a comprehensive research analysis of data set and use it for prediction of post operative complications in a subset of pancreaticoduodenectomy cases. The different models formulated and tested are based on sound reasoning and parameters. I hope that, with use of similar computer-generated applications in health care research would help resolve many issues in future

6. PLOS authors have the option to publish the peer review history of their article (what does this mean?). If published, this will include your full peer review and any attached files.

Reviewer #1: **Yes: **sorayya malek

Reviewer #2: No

---

## [Author Response · Author response to Decision Letter 0]

7 Jun 2024

Dear Dr. Chen

Thank you for the opportunity to resubmit our manuscript. Please find a point-by-point response to reviewers’ comments below. Also, please do not hesitate to reach out to us if further information is required. 

On behalf of the authors

Martin Sillesen MD, PhD

Assoc. Prof. of Surgery

Dep. of Organ Surgery and Transplantation

Copenhagen University Hospital, Rigshospitalet

Denmark

Martin.Sillesen@Regionh.dk

Journal Requirements:

Comment #1

Response:

Thank you for highlighting this. The formatting issues have been addressed. 

Comment #2

Response:

Thank you for this question. The project received a waiver of IRB approval from the Massachusetts General Hospital IRB. This is stated in the methods section. We have expanded this section further to detail that this entailed that patient consent was not required and thus not obtained. 

Comment #3

3. Thank you for stating the following in the Competing Interests section: "I have read the journal's policy and the authors of this manuscript have the following competing interests: Authors AB and MS have founded Aiomic Aps, a healthtech company fielding artificial intelligence models for healthcare use. The present work is for research only and is not related to any commercial activities."

Response:

You are welcome

Comment #4

Response:

Thank you, we have updated the cover letter with the required information. 

Reviewers' comments:

Reviewer's Responses to Questions

Comments to the Author

Reviewer #1: 

Comment #5

I appreciate the opportunity to examine this interesting paper. Nevertheless, there are a few concerns regarding the work.

Conduct a comprehensive review of existing literature pertaining to the utilization of machine learning and deep learning techniques in the context of mortality and morbidity, as well as any relevant studies in related fields. The utilization of Deep Neural Networks (DNN) and Random Forest (RF) can be justified based on their respective advantages and capabilities. There are numerous AI algorithms available, with the latest one being ensemble learning, which offers superior performance. Deep neural networks (DNN) may not consistently perform as effectively as other machine learning methods, but they are particularly notable for their application in image processing. This is evident when the area under the curve (AUC) has a relatively low value. A value that is considered acceptable is 0.75 or higher.

The authors should explicitly state their objective and anticipated outcome in the objective section. As I comprehend, there are multiple morbidity outcomes and death outcomes, and the authors conducted different modeling for each result. Alternatively, is it a prediction involving many classes? The authors should explicitly articulate this.

Response: 

Thank you these comments. Overall, we fully agree with the reviewer that there are multiple different modeling options well suited for this prediction task. We have chosen the DNN architecture, as we have previously demonstrated superior performance using this approach on a general surgical dataset (Manuscript reference #12). 

In this modeling study, we use the DNNs to do multiclass prediction, so it is essentially the same network used for multi-label prediction of several different postoperative complications. We have augmented the manuscript with a further description of this important point, as well as elaborated on the expected outcome in the introduction as requested. 

Comment #6

In the data section, specify the percentage of missing values for final variables and identify the data imputation method employed.

Response :

Thank you for this question. We agree that the issue of missing data is important. For the final (output) variables, the dataset was complete and thus did not have missing data. This is because the NSQIP database does not allow for incomplete postoperative complication data entry when submitting cases. We have added this information to the methods section of the manuscript, as well as included a description of how missing data was handled for input variables. For the input data, an overview of demographic variables between datasets, as well as the percentage of missing data, has now been included in the supplementary data table 1. 

Comment #7

The authors determined the final features from the 150 variables listed in the study. The ultimate result must also be explicitly expressed. It is necessary to provide a clear explanation of how the dataset is allocated to the four models. How did the authors address the issue of data imbalance, which can explain the low AUC value? There is a noticeable imbalance in the data when merging patients with PD with people without PD.

Response:

Thank you for this insightful question. We agree that a full overview of the input features is important for assessing the data on which the model performs its predictions. We have added a table detailing the input features as supplementary data. 

The reviewer is indeed correct in stating that the class imbalance in the data may well impact on the AUC performance score. This problem often occurs in postoperative complication data, as the outcomes are infrequent. We have, however, sought to address this through the introduction of positive weights applied to the loss function of the DNN. This approach has been further detailed in the methods section

Comment #8

Summary statistics of each dataset are compared against the outcome using chi square analysis, yielding the chi square value for categorical and corresponding p-value continuous variable.

Response:

In this study, we have not calculated summary statistics of the different datasets for several reasons. First, the objective is not to determine whether significant differences existed between input and output variables. This is no doubt the case, as a general surgical cohort will for example have a different spectrum and incidence of postoperative complications as would a specific pancreatic cohort. We thus feel that the objective of the study is not to compare cohorts and identify differences in these, but rather to investigate how optimal training of a PD specific model can be achieved. 

Even if it was chosen that summary statistics were required, the number of input and output variables would necessitate correction for multiple testing due to the number of variables included. As such, there would be a clear risk of a Type-II error, which again would diminish the usefulness of such a comparison. While this can in certain cases of cause be acceptable, we respectfully feel that a comparison of cohorts is not within the objective of this study. Rather this is a model comparison study. 

Comment #9

Did the authors undertake parameter optimization on the RF model to enhance its performance? Support Vector Machines (SVM) are a suitable option for conducting benchmarking.

Response:

Thank you for this comment. We agree that the SVM approach could be useful a study such as this. As suggested, we have benchmarked the RF models with an SVM approach with parameter optimization using grid search. This approach was inferior to the RF approach, resulting in a ROC AUC of 0.52 for morbidity and 0.50 for mortality. As such, the RF outperformed the grid search optimized SVM approach, indicating adequate hyper parameter optimization for the RF approach. This information has been added to the manuscript. 

Comment #10

I suggest incorporating Net Index Reclassification (NRI) for the purpose of comparing it with traditional methodologies.

Response:

Thank you for this suggestion. As suggested, we have included the NRI calculations as supplementary data, and updated the manuscript accordingly.

Comment #11

Additionally, the use of SHAP can provide a clearer understanding of the relationship between input properties and the resulting conclusion.

Response:

Thank you for this suggestion. We agree that the addition of SHAP values would add value to the manuscript. We have thus included these in the revised manuscript. 

Comment#12

Reviewer #2: I commend you for taking up a comprehensive research analysis of data set and use it for prediction of post operative complications in a subset of pancreaticoduodenectomy cases. The different models formulated and tested are based on sound reasoning and parameters. I hope that, with use of similar computer-generated applications in health care research would help resolve many issues in future

Response:

Thank you for this comment!

---

## [Decision Letter · Decision Letter 1]

21 Oct 2024

PONE-D-23-30439R1ASSESSING THE VALUE OF DEEP NEURAL NETWORKS FOR POSTOPERATIVE COMPLICATION PREDICTION IN PANCREATICODUODENECTOMY PATIENTSPLOS ONE

Dear Dr. Sillesen,

Thank you for submitting your manuscript to PLOS ONE. After careful consideration, we feel that it has merit but does not fully meet PLOS ONE’s publication criteria as it currently stands. Therefore, we invite you to submit a revised version of the manuscript that addresses the points raised during the review process.

Please address the preprint issue.

We look forward to receiving your revised manuscript.

Kind regards,

Academic Editor

PLOS ONE

Reviewers' comments:

Reviewer's Responses to Questions

**Comments to the Author**

1. If the authors have adequately addressed your comments raised in a previous round of review and you feel that this manuscript is now acceptable for publication, you may indicate that here to bypass the “Comments to the Author” section, enter your conflict of interest statement in the “Confidential to Editor” section, and submit your "Accept" recommendation.

Reviewer #2: All comments have been addressed

Reviewer #3: All comments have been addressed

2. Is the manuscript technically sound, and do the data support the conclusions?

Reviewer #2: Yes

Reviewer #3: Partly

3. Has the statistical analysis been performed appropriately and rigorously? 

Reviewer #2: Yes

Reviewer #3: I Don't Know

4. Have the authors made all data underlying the findings in their manuscript fully available?

Reviewer #2: Yes

Reviewer #3: No

5. Is the manuscript presented in an intelligible fashion and written in standard English?

Reviewer #2: Yes

Reviewer #3: No

6. Review Comments to the Author

Reviewer #2: Thank you very much for addressing the reviewers queries and concerns. I believe it would be suitable in the updated form.

Reviewer #3: This is a preprint paper from

https://www.medrxiv.org/content/10.1101/2023.08.21.23294364v1

This is a preprint paper from

https://www.medrxiv.org/content/10.1101/2023.08.21.23294364v1

7. PLOS authors have the option to publish the peer review history of their article (what does this mean?). If published, this will include your full peer review and any attached files.

Reviewer #2: No

Reviewer #3: **Yes: **hazim alhiti

---

## [Author Response · Author response to Decision Letter 1]

8 Nov 2024

Dear PLOSOne editorial team

Thank you for assessing our submitted manuscript. We understand that reviews feel that questions have been addressed, but here seems to be an issue with the fact that the manuscript has previously been submitted as a preprint. 

To formally adhere to the review phases, please find a point-by-point response to comments below. 

Thank you again for the opportunity to resubmit our manuscript. 

On behalf of the authors

Martin Sillesen MD, PhD

Assoc. Prof. of Surgery

Dep. of Organ Surgery and Transplantation

Copenhagen University Hospital, Rigshospitalet

Denmark

Martin.Sillesen@Regionh.dk

Reviewers' comments:

Reviewer's Responses to Questions

Comment #1

Comments to the Author

1. If the authors have adequately addressed your comments raised in a previous round of review and you feel that this manuscript is now acceptable for publication, you may indicate that here to bypass the “Comments to the Author” section, enter your conflict of interest statement in the “Confidential to Editor” section, and submit your "Accept" recommendation.

Reviewer #2: All comments have been addressed

Reviewer #3: All comments have been addressed

Response: 

Thank you

Comment #2

6. Review Comments to the Author

Reviewer #2: Thank you very much for addressing the reviewers queries and concerns. I believe it would be suitable in the updated form.

Reviewer #3: This is a preprint paper from

https://www.medrxiv.org/content/10.1101/2023.08.21.23294364v1

This is a preprint paper from

https://www.medrxiv.org/content/10.1101/2023.08.21.23294364v1

Response:

It is indeed true that this manuscript has previously been submitted as a preprint, which we believe is in accordance with PLOSOne guidelines. This seems to be more of a comment pertaining to the fact this this has been presented as a preprint, rather than a request for edits from the side of the reviewer. 

However, from conversations with the editorial office, we understand that there is a request from the editorial office for “…authors to make more improvements from the version in the preprint. Please revise to show more improvements.”. 

The original preprint version was submitted in August 2023, whereas the present version was substantially edited based on the reviewer’s comments. As such, the preprint and the current manuscript is less than 80% identical to the preprint, indicating that 20% of the text has been updated/altered compared with the preprint. 

Furthermore, based on the reviewers inputs, we have updated the manuscript with new tables and data, added new modelling approaches including benchmarking with SVMs and included parameter optimization . We have, as per reviewers suggestions, included Net Index Classification and Shapley insights into driving factors as well as performed a general editing of the manuscript text. 

As such, we would respectfully argue that the current version of the manuscript is indeed very different from the preprint version. 

While we are of course open to performing other changes to the manuscript, we find it difficult to make changes to a manuscript that is already significantly different from its preprint version both in terms of text, data, figures, tables and methods without more specific instructions on where the editors see the need for improvements specifically.

---

## [Decision Letter · Decision Letter 2]

10 Dec 2024

ASSESSING THE VALUE OF DEEP NEURAL NETWORKS FOR POSTOPERATIVE COMPLICATION PREDICTION IN PANCREATICODUODENECTOMY PATIENTS

PONE-D-23-30439R2

Dear Dr. Sillesen,

We’re pleased to inform you that your manuscript has been judged scientifically suitable for publication and will be formally accepted for publication once it meets all outstanding technical requirements.

Kind regards,

Robert Jeenchen Chen, MD, MPH, ChFC®, EA, CLU

Academic Editor

PLOS ONE

Additional Editor Comments (optional):

Reviewers' comments:

Reviewer's Responses to Questions

**Comments to the Author**

1. If the authors have adequately addressed your comments raised in a previous round of review and you feel that this manuscript is now acceptable for publication, you may indicate that here to bypass the “Comments to the Author” section, enter your conflict of interest statement in the “Confidential to Editor” section, and submit your "Accept" recommendation.

Reviewer #3: All comments have been addressed

2. Is the manuscript technically sound, and do the data support the conclusions?

Reviewer #3: (No Response)

3. Has the statistical analysis been performed appropriately and rigorously? 

Reviewer #3: Yes

4. Have the authors made all data underlying the findings in their manuscript fully available?

Reviewer #3: Yes

5. Is the manuscript presented in an intelligible fashion and written in standard English?

Reviewer #3: Yes

6. Review Comments to the Author

Reviewer #3: hard work

well done

congratulations

my final decision is acceptable ‎

my final decision is acceptable ‎

7. PLOS authors have the option to publish the peer review history of their article (what does this mean?). If published, this will include your full peer review and any attached files.

Reviewer #3: **Yes: **Hazim Alhiti

---

## [Editor Report · Acceptance letter]

15 Dec 2024

PONE-D-23-30439R2 

PLOS ONE

Dear Dr. Sillesen, 

I'm pleased to inform you that your manuscript has been deemed suitable for publication in PLOS ONE. Congratulations! Your manuscript is now being handed over to our production team.

Kind regards, 

on behalf of

Dr. Robert Jeenchen Chen 

Academic Editor

PLOS ONE